# Cost-utility analysis of adding abiraterone acetate plus prednisone/prednisolone to long-term hormone therapy in newly diagnosed advanced prostate cancer in England: Lifetime decision model based on STAMPEDE trial data

Caroline S. Clarke[1]*, Rachael M. Hunter[1], Andrea Gabrio[2], Christopher D. Brawley[3], Fiona C. Ingleby[3,4], David P. Dearnaley[5], David Matheson[6], Gerhardt Attard[7], Hannah L. Rush[3,8], Rob J. Jones[9,10], William Cross[11], Chris Parker[12], J. Martin Russell[9,10], Robin Millman[13], Silke Gillessen[14,15,16], Zafar Malik[17], Jason F. Lester[18], James Wylie[19], Noel W. Clarke[19,20], Mahesh K. B. Parmar[3], Matthew R. Sydes[3], Nicholas D. James[3,5]

1 Research Department of Primary Care and Population Health, University College London, London, United Kingdom, 2 Department of Methodology and Statistics, Faculty of Health Medicine and Life Sciences, Maastricht University, Maastricht, Netherlands, 3 MRC Clinical Trials Unit at UCL, Institute of Clinical Trials and Methodology, University College London, London, United Kingdom, 4 Faculty of Epidemiology and Population Health, London School of Hygiene and Tropical Medicine, London, United Kingdom, 5 Institute of Cancer Research and The Royal Marsden NHS Foundation Trust, London, United Kingdom, 6 Patient Representative, University of Wolverhampton, Wolverhampton, United Kingdom, 7 University College London Cancer Institute, London, United Kingdom, 8 Guys and St Thomas' NHS Foundation Trust, London, United Kingdom, 9 Beatson West of Scotland Cancer Centre, Glasgow, United Kingdom, 10 Institute of Cancer Sciences, University of Glasgow, Glasgow, United Kingdom, 11 Department of Urology, Leeds Teaching Hospitals NHS Trust, Leeds, United Kingdom, 12 Royal Marsden Hospital and Institute of Cancer Research, Sutton, United Kingdom, 13 Patient Representative, MRC Clinical Trials Unit at UCL, Institute of Clinical Trials and Methodology, University College London, London, United Kingdom, 14 Division of Cancer Sciences, University of Manchester, Manchester, United Kingdom, 15 Oncology Institute of Southern Switzerland, EOC, Bellinzona, Switzerland, 16 Università della Svizzera Italiana, Lugano, Switzerland, 17 Clatterbridge Cancer Centre NHS Foundation Trust, Birkenhead, United Kingdom, 18 South West Wales Cancer Centre, Singleton Hospital, Swansea, United Kingdom, 19 Christie NHS Foundation Trust, Manchester, United Kingdom, 20 Salford Royal Hospital, Salford, United Kingdom

* caroline.clarke@ucl.ac.uk

## Abstract

Adding abiraterone acetate (AA) plus prednisolone (P) to standard of care (SOC) improves survival in newly diagnosed advanced prostate cancer (PC) patients starting hormone therapy. Our objective was to determine the value for money to the English National Health Service (NHS) of adding AAP to SOC. We used a decision analytic model to evaluate cost-effectiveness of providing AAP in the English NHS. Between 2011–2014, the STAMPEDE trial recruited 1917 men with high-risk localised, locally advanced, recurrent or metastatic PC starting first-line androgen-deprivation therapy (ADT), and they were randomised to receive SOC plus AAP, or SOC alone. Lifetime costs and quality-adjusted life-years (QALYs) were estimated using STAMPEDE trial data supplemented with literature data where necessary, adjusting for baseline patient and disease characteristics. British National

**Data Availability Statement:** The STAMPEDE study involves human research participants and contains sensitive information. We therefore will make the dataset available on request via our established MRC CTU at UCL processes, described here: (https://www.mrcctu.ucl.ac.uk/our-research/other-research-policy/data-sharing/).

**Funding:** This cost-effectiveness analysis was supported by Cancer Research UK (https://www.cancerresearchuk.org/) (awarded to CSC and RMH) and the MRC Clinical Trials Unit at UCL (https://www.mrcctu.ucl.ac.uk) as an add-on to CRUK/06/019 (awarded to NDJ). STAMPEDE grant codes: CRUK_A12459; Medical Research Council (https://mrc.ukri.org), MRC_MC_UU_12023/25 (awarded to NDJ). STAMPEDE is registered on http://www.clinicaltrials.gov (NCT00268476; first posted: 22 December 2005) (http://www.stampedetrial.org). The funders had no role in study design, data collection and analysis, decision to publish, or preparation of the manuscript.

**Competing interests:** Besides the funding declared above, • CDB reports grants from Novartis, Sanofi-Aventis, Pfizer, Janssen Pharma, Cancer Research UK, and Medical Research Council. • DD reports personal fees from The Institute of Cancer Research, grants from Cancer Research UK Program Grant and personal fees from Janssen. In addition, DD has a patent EP1933709B1 issued. • GA reports receiving commercial research grants from Janssen and AstraZeneca; has received honoraria and/or travel support from the speakers' bureaus of Janssen, Astellas, Pfizer, Ferring, Sanofi-Aventis and Roche/Ventana; and has served as a consultant for/advisory board member of Janssen, Bayer, Astellas, Pfizer, Novartis, AstraZeneca, Orion, and Essa. GA has an ownership interest (including patents) in The Institute of Cancer Research Rewards to Discoverers for abiraterone acetate. • RJJ reports grants and personal fees from Astellas, AstraZeneca, Exelixis, and Roche; grants, personal fees and non-financial support from Bayer; personal fees and non-financial support from Bristol Myers Squibb, Janssen, Ipsen, and MSD; and personal fees from Merck Serono, Novartis, Pfizer, Sanofi Genzyme, and EUSA. • WC reports grants from Myriad Genetics. • CP reports personal fees from Bayer, Clarity, ITM (Isotopen Technologien Muenchen AG), Janssen, and Myovant. • SG reports personal fees from Sanofi, Orion, Roche, Janssen Cilag, and Amgen; other benefits from Menarini Silicon Biosystems, Bayer, AAA International, ProteoMediX, Toledo, and MSD; personal fees and other benefits from Astellas Pharma; and grants from Astellas Pharma. • ZM

Formulary (BNF) prices (£98/day) were applied for AAP. Costs and outcomes were discounted at 3.5%/year. AAP was not cost-effective. The incremental cost-effectiveness ratio (ICER) was £149,748/QALY gained in the non-metastatic (M0) subgroup, with 2.4% probability of being cost-effective at NICE's £30,000/QALY threshold; and the metastatic (M1) subgroup had an ICER of £47,503/QALY gained, with 12.0% probability of being cost-effective. Scenario analysis suggested AAP could be cost-effective in M1 patients if priced below £62/day, or below £28/day in the M0 subgroup. AAP could dominate SOC in the M0 subgroup with price below £11/day. AAP is effective for non-metastatic and metastatic disease but is not cost-effective when using the BNF price. AAP currently only has UK approval for use in a subset of M1 patients. The actual price currently paid by the English NHS for abiraterone acetate is unknown. Broadening AAP's indication and having a daily cost below the thresholds described above is recommended, given AAP improves survival in both subgroups and its cost-saving potential in M0 subgroup.

# Introduction

Prostate cancer is the most common cancer in men in the UK [1], and second most common cancer in men worldwide [2]. Long-term hormone therapy has been first-line standard of care for locally advanced and metastatic prostate cancer since the 1960s [3]. Disease progression occurring during treatment with long-term hormone therapy represents a transition to a disease state referred to as castration-resistant prostate cancer (CRPC), and recent modifications made to the treatment pathway after CRPC onset have led to survival and morbidity improvements [4, 5]. Additions of recently approved CRPC therapies to long-term hormone therapy in pre-CRPC, or 'hormone-naïve', patients are being tested via several trials including the STAMPEDE trial [5–10] which began in 2005. It uses a multi-arm, multi-stage design, allowing testing of emerging single and combination treatments against one, continuous standard-of-care arm based on long-term androgen-deprivation therapy (ADT) [11]. STAMPEDE is registered at clinicaltrials.gov (NCT00268476).

STAMPEDE showed that adding abiraterone acetate plus prednisone/prednisolone (AAP) to ADT in first line resulted in significantly improved overall and failure-free survival compared to ADT alone, with a 3-year survival rate of 83% in AAP plus ADT compared to 76% for ADT only [12] in a broader population than that covered by the current AAP indication in the UK [5, 13]. The administration of AAP is however associated with significant financial cost. In England, approval of new technologies for National Health Service (NHS) implementation requires evidence of sufficient gains in quality-adjusted life-years (QALYs) to justify additional cost [14] If a new intervention carries additional cost over and above the cost to the health system of generating a new QALY, then this implies displacement or loss of health elsewhere in the system [15]. It is not clear whether the balance of health gains and extra cost associated with the addition of AAP is acceptable in this context. The cost-effectiveness of AAP in comparison to prednisolone as best supportive care, in metastatic CRPC patients previously treated with docetaxel, has previously been explored as part of UK National Institute for Health and Care Excellence (NICE) submissions [16]. This paper describes the first cost-effectiveness analysis (CEA) comparing AAP+ADT to ADT alone as first-line treatments in the newly diagnosed population, directly from trial-based data. Other work has reported CEAs in this population using aggregated data [17, 18].

reports involvement in consultancy and advisory boards at Janssen and Sanofi, in advisory boards at Astellas, and sponsorship to attend medical conferences from Astellas, Bayer and Janssen. • NWC reports receiving research grants AstraZeneca and Janssen; honoraria and/or travel support from the speakers' bureaus of Janssen, Astellas, Ferring, Sanofi-Aventis and has served as a consultant for/advisory board member of Janssen, Bayer, Astellas, Ferring, and AstraZeneca. • MKBP reports unrestricted grant funding to contribute to STAMPEDE overall from Astellas, Clovis Oncology, Novartis, Pfizer and Sanofi. • MRS reports grants from Clovis, grants and non-financial support from Astellas, Janssen, Novartis, Pfizer, and Sanofi to support the running of STAMPEDE; and personal fees from Lilly Oncology and Janssen for educational events unconnected to the submitted work or the underpinning trial. • NDJ reports grants and personal fees from Janssen, Astellas, and Sanofi, and personal fees from AstraZeneca, during the conduct of the study. • CSC, RMH, AG, FCI, DM, HLR, JMR, RM, JFL and JW have nothing to disclose. This does not alter our adherence to PLOS ONE policies on sharing data and materials.

The aim of this paper was to evaluate the cost-effectiveness of AAP in men initiating long-term ADT for prostate cancer. We report the results of a lifetime cost-utility analysis (CUA) based on short-term trial-based patient-level data on healthcare resource use, health-related quality of life (EQ-5D-3L), disease progression and mortality collected in the STAMPEDE trial. The lifetime model extrapolates from trial-based survival analyses, calculating survival, discounted quality-adjusted survival and costs, over a lifetime (45-year) horizon.

## Methods

### STAMPEDE trial—"Abiraterone comparison"

Patients recruited between November 2011 and January 2014 and randomly allocated to AAP plus standard of care (SOC) based on ADT, or SOC alone, (1:1 allocation) constituted STAMPEDE's "abiraterone comparison". All trial participants provided written informed consent. The STAMPEDE trial was conducted in accordance with the principles of Good Clinical Practice guidelines and the Declaration of Helsinki, and the appropriate regulatory and ethics approvals were obtained for the original study and all amendments (http://www.stampedetrial.org/centres/essential-documents/ethics-regulatory/). No additional ethics approval was required for the cost-utility analysis reported here.

The database was locked on 10 February 2017. Those eligible for randomisation had newly diagnosed and metastatic, node-positive, or high-risk locally advanced, non-metastatic prostate cancer, or disease previously treated with radical surgery or radiotherapy which was relapsing with certain high-risk features. There were no age constraints, but patients with known severe cardiovascular disease were excluded from the trial. Further details on the rationale and design of STAMPEDE are described in the protocol and other trial documentation [6, 11, 12, 19, 20].

Treatments in both arms included ADT for at least 2 years. For patients randomised to AAP+SOC, AAP (abiraterone acetate 1000mg, prednisolone 5mg) was given once daily, with duration dependent on disease stage. Patients with metastatic disease at baseline were offered AAP treatment until progression whereas most non-metastatic patients only received it for 2 years. Information was collected on administration of AAP and other medications including hormone therapy (including ADT), chemotherapy, bisphosphonates, radioisotopes, steroids, pain medications, and other cancer medications; surgical and other procedures; radiotherapy; and unscheduled primary care and inpatient and outpatient secondary care visits; as well as on deaths and serious adverse events (SAEs). The EuroQol EQ-5D 3-level (EQ-5D-3L) questionnaire was administered at baseline and each trial visit. Trial visits were every 6 weeks up to 6 months, then every 12 weeks up to 2 years, then every 6 months up to 5 years, or until progression.

Results are reported separately in this paper for (i) patients who initially presented without metastases, or with only lymph node metastasis as this tends to be underdiagnosed in practice and treatment pathways as well as prognosis tend to be similar across these patients (hereafter referred to as M0 subgroup), and (ii) those who presented with other metastatic disease (hereafter M1 subgroup), due to the different treatment pathways and prognoses associated with each group [21].

### Patient and public involvement statement

Patients and the public are involved in the ongoing management of the STAMPEDE study itself, but were not separately involved in this health economic analysis.

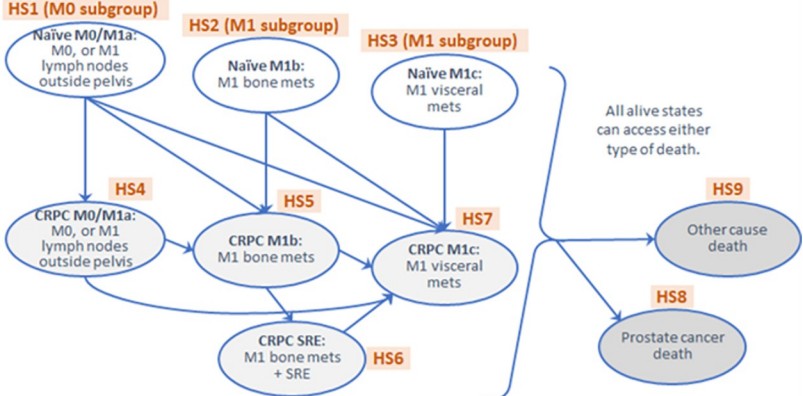

**Fig 1. Model structure.** Nine health states with 25 allowed transitions (11 arrows among HS1-HS7, plus 1 transition from each alive health state (HS1-7) to each dead health state (HS8-9)).

## Trial-based analysis

**Survival analysis.** A trial-based analysis was performed to describe the trial data and identify additional information required for the lifetime simulation model to describe the real-life pathway and account for bias due to short trial follow-up. Each patient's status during the trial was categorised as one of 9 health states (HS) that matched those used in the previously published docetaxel analysis from STAMPEDE [22] (Fig 1). The three health states which covered the trial's eligibility criteria were: (i) non-metastatic disease or with lymph-node metastasis (HS1), (ii) bone metastases (HS2), and (iii) visceral metastases (HS3). These distinctions were made on the basis of previously observed prognostic differences between the three groups [23, 24]. Patients in HS1 comprised the non-metastatic (M0) subgroup, and patients in HS2 and HS3 the metastatic subgroup (M1). After treatment failure, patients were considered to be in CRPC health states split according to the same groups as above (HS4, HS5 and HS7, respectively), with an additional state (HS6) where skeletal-related events (SRE) were observed after bone metastasis. SREs occurring in treatment-naïve patients (HS1) without a metastasis event were considered as treatment failure; these patients were moved to HS4 and not HS6. Information was also collected on patient deaths and whether they were prostate cancer related or not (HS8 and HS9). Participants alive at the end of trial follow-up were censored at last documented follow-up.

Parametric survival curves were modelled jointly or separately for the 25 transitions in Fig 1 using trial data [25–28], to estimate lifetime event rates in the lifetime model described in the *Methods*: *Lifetime simulation model* section, adjusting for a range of baseline variables (S1 File). The first transition by a patient represented failure-free survival as defined in the clinical analysis [12] so the time to this first event was estimated jointly for the 9 allowed transitions from any of HS1/2/3 to any of HS4/5/7/8, as a function of randomised group and certain prognostic baseline patient and disease characteristics [23, 24]. This model used a flexible parametric model with 5 degrees of freedom, similarly to the clinical analysis [12], and the best fit was on the log cumulative odds scale. The 7 transitions to other-cause death (HS1-7 to HS9) were estimated jointly as were dependent on age. The remaining transitions were estimated separately or jointly depending on event numbers and convergence, as a function of randomised group and time from randomisation to failure, adjusting for baseline patient and disease characteristics.

Two significant treatment pathway changes occurred during the trial analysis timeframe, and baseline variables marking these were assessed for inclusion in the survival models. Firstly, NICE approved use of AAP and enzalutamide for patients who had received prior chemotherapy in 2012, and this was extended to all patients at first relapse in 2016 [29], but had been available earlier in England via the Cancer Drugs Fund, therefore a binary variable indicating recruitment location as England or elsewhere was included in the analysis. Secondly, radiotherapy (RT) was removed from SOC for newly diagnosed M1 patients from January 2013 when the next study arm added to STAMPEDE opened, so a binary variable indicating recruitment time period was included.

Exponential, generalised gamma, Weibull and lognormal models were assessed for all individual and joint models, as well as splines for some models where convergence or good fit was not achieved from these initial approaches. Goodness of fit was assessed using the Akaike Information Criterion (AIC) [30] as well as visual inspection comparing the observed Kaplan-Meier curve with the predicted within-trial survival curve for each transition, and longer-term plausibility of extrapolations also considered.

**Utility scores.** Utility scores were calculated from complete responses to trial EQ-5D-3L questionnaires using the standard UK tariff [31]. Partially completed questionnaires were deemed missing [32]. Multiple imputation using chained equations (MICE) and predictive mean matching [33] was implemented in R to impute missing utility scores by arm, based on five imputations and 30 iterations [32]. Convergence checks revealed no apparent issues in the algorithm. The imputation model included all potentially prognostic variables available at baseline, initial trial eligibility category, randomised group, timing of quality-of-life data collection, quality-of-life data over time, health state, and death during the trial. Utility was imputed as zero from date of death. As utility scores are negatively skewed with a ceiling effect (high proportion of patients reporting perfect utility of 1), two-part regression was performed on the imputed datasets, where a logit model (binomial family with log link) predicted the likelihood of utility being equal to 1, then a gamma model (gamma family with identity link) predicted the utility score if it differed from 1. Gamma models allow values greater than zero, so values predicted were one minus the utility score. Regression models included baseline characteristics considered predictive of utility based on clinical opinion (age, World Health Organization (WHO) performance status, nodal stage), health state (Fig 1), and a time-dependent version of the randomised group, as the impact of randomised group on quality of life was assumed to last for the first year only, leading to a three-category "treatment in first year" parameter: (i) first year AAP+SOC, (ii) first year SOC-only, or (iii) second year onwards either group. The best overall fit was chosen according to the Quasi-likelihood Information Criterion (QIC) [34] and the final model's overall appropriateness was assessed using the Basu-Manca test series [35].

**Costs.** Costs were calculated based on healthcare resource use, using the English NHS perspective because the majority of patients were recruited from this nation, in 2017–18 £ prices. Cost information from trial data was analysed in three categories: (i) investigational medications; (ii) other specific (expensive) medications (docetaxel, enzalutamide, cabazitaxel and radium); and (iii) general disease management costs (other medications, procedures, unscheduled visits, radiotherapy). Information on medications, procedures and therapies including radiotherapy was captured during the trial as start and stop dates, and doses and frequencies. Changes were made to SOC during the course of the study, meaning that some patients in the SOC-only arm also received AAP at some point, mostly during later disease stages. All dose information for AAP in the SOC-only arm was however missing, so was imputed as 1000mg/day, as this was both the indicated and the modal observed amount. Missing prednisolone amounts were imputed assuming protocol dose (5mg/day) and that start/stop dates matched those of abiraterone. Other missing medication dose information was imputed as modal

observed or British National Formulary (BNF) indicated doses due to insufficient information for informative multiple imputation. Unit costs were obtained from standard sources, including the BNF [36], NHS Reference Costs 2017–18 [37], and the published docetaxel CUA [22], adjusting where necessary to 2017–18 prices using the new Health Services Index using CPI (Consumer Price Index) (Health) and the previous Hospital and Community Health Services indices [37, 38]. The base-case cost for abiraterone was the BNF cost (£97.68/day). The NHS purchases abiraterone acetate at an undisclosed discount, so a threshold analysis explored the impact of using lower prices (see *Methods*: *Overall incremental cost-effectiveness and sensitivity analysis* section). Reductions of around 20% were made to BNF costs for enzalutamide, cabazitaxel and radium to better reflect prices paid by the NHS [22]. As 6-week general disease management costs were positively skewed, a two-stage regression for costs below and above a boundary amount per 42-day cycle was performed, by arm and health state, controlling for baseline age group, WHO status, nodal stage, and the three-category variable, 'treatment in first year'. Generalised gamma and Gaussian models were assessed, with models chosen according to QIC, including exploring the position of the boundary amount separating the two parts of the regression. Unit costs and other further details are given in S2 File.

### Lifetime simulation model

**Simulation model structure.** The lifetime model was a patient-level simulation Markov model, performed in R (v3.6.3) [39] using standard packages and additional functions written by Woods et al. [22]. The model generated lifetime information on time patients spent in each state, by arm and subgroup, using a 42-day cycle length and based on the survival models calculated from trial data described in *Methods*: *Trial-based analysis*: *Survival analysis* section. Office for National Statistics (ONS) life tables [40] were used (Gompertz distribution) instead of other-cause death parametric survival model from the trial data if the predicted date of death was after the participant's last follow-up. The time horizon was 45 years after randomisation, since the mean age of the youngest category was 55 years, therefore likely capturing all patients' lifetimes.

Given the limited information on outcomes beyond onset of CRPC available via STAMPEDE, survival estimations for M0 patients were applied based on data from M1 patients who had progressed to CRPC disease [41]. Forty simulations generated per patient profile provided stable results in the deterministic analyses, and 25 in the probabilistic sensitivity analyses (500 iterations). The latter were used to provide points on the cost-effectiveness plane that were then translated onto the cost-effectiveness acceptability curve [42] and into 95% confidence intervals for costs and QALYs per arm and subgroup.

**Utility scores and quality-adjusted life-years.** Parameters from the utility regression described in the *Methods*: *Trial-based analysis*: *Utility scores* section were combined with time-in-state information from the lifetime simulation described in the *Methods*: *Lifetime simulation model*: *Simulation model structure* section to give overall per-patient lifetime QALYs by arm and subgroup. Area-under-the-curve methods were used to calculate QALYs [43], and future QALYs were discounted at 3.5% per year [14].

**Costs.** Parameters from the trial cost analyses described in the *Methods*: *Trial-based analysis*: *Costs* section were combined with time-in-state information from the lifetime simulation described in the *Methods*: *Lifetime simulation model*: *Simulation model structure* section and additional information to give overall per-patient lifetime costs by arm and subgroup, with future costs discounted at 3.5% per year [14]. Costs for the group of specific medications were estimated according to health state and whether they had been in that state for (i) less than a year, (ii) 1–2 years, or (iii) more than 2 years, in order to provide optimal sensitivity in the analysis for these more expensive drugs when moving from trial-based results to the longer

time horizon in the lifetime model where costs and outcomes were applied in cycles. AAP costs were estimated in the same way, and also split by randomised group.

Additional information on top of trial costs included a flat SAE cost calculated using SAE information collected in the trial. Published costs for end-of-life care in prostate cancer were also included (£6,897, adjusted to 2017–18 prices from Round et al.) [44], as were estimated costs for standard monitoring activities, and stoppage of medications where this implied additional healthcare resources. Unit costs and other further details are given in S3 File.

## Overall incremental cost-effectiveness and sensitivity analysis

The overall outputs of the lifetime model are presented by subgroup (M0 and M1) as the undiscounted incremental survival, the discounted incremental QALYs, the discounted incremental costs (£), and the incremental cost per QALY gained. One-way deterministic sensitivity analysis was performed as a threshold analysis to explore the impact of using a lower abiraterone acetate input price, using 75%, 50%, 25% and 10% of the base-case full BNF price, as the BNF price was likely an over-estimate of current price paid by the NHS. Joint probabilistic sensitivity analysis was performed to explore the impact of uncertainties in model inputs. Five hundred sets of parameters from the trial analysis regressions (2-part regression for utility scores, 2-part regression for general disease management costs) were generated, using multivariate normal distributions described by the calculated regression parameters. Costs for investigational medications, other specific medications, monitoring, SAE premiums and end-of-life one-off costs were varied according to gamma distributions with random deviates. The results were plotted on cost-effectiveness planes and used to draw cost-effectiveness acceptability curves, to show the likelihood of AAP+SOC arm being more cost-effective than SOC-only arm at various cost-effectiveness thresholds.

This analysis is reported according to the Consolidated Health Economic Evaluation Reporting Standards (CHEERS) statement [45] and follows the NICE reference case [14]. Analysis was performed in R (version 3.6.3) [39] with some preparatory work performed in Stata v14 [46] and v16 [47].

## Results

### Headline results from overall lifetime model

The discounted per-patient lifetime gains in QALYs were 0.33 QALYs for M0 patients and 1.48 QALYs for M1 patients according to the base-case deterministic simulation analysis. Following the AAP+SOC treatment pathway cost an additional £48,821 per patient in the M0 subgroup, and £70,246 in M1 subgroup, compared to SOC alone. The base-case ICERs were therefore £149,748 per QALY gained in the M0 subgroup and £47,503 per QALY gained in the M1 subgroup (Table 1). The cost-effectiveness of AAP+SOC compared to SOC-only was above the thresholds used in England (£13,000 to £30,000/QALY gained) [14, 15] when using the BNF price for abiraterone acetate and therefore not cost-effective. Further details and breakdowns are given in S8 and S9 Files for the probabilistic sensitivity analysis results.

The rest of the Results section here discusses results from the trial-based analysis that were required to feed into the lifetime model, and then goes on to describe the survival results of the lifetime model and its various sensitivity analyses.

### Trial-based results

**Trial population.**   1917 patients at 111 United Kingdom (UK) and 5 Swiss sites were randomised to AAP+SOC (n = 960) or SOC-only (n = 957). A full description of the patient

**Table 1. Total lifetime per-patient costs and QALYs split by arm and subgroup, calculated using the deterministic base-case simulation analysis, and corresponding ICERs by subgroup.**

| | M0 subgroup | | | M1 subgroup | | |
|---|---|---|---|---|---|---|
| | AAP+SOC | SOC-only | Difference | AAP+SOC | SOC-only | Difference |
| Lifetime costs | 97,558 | 48,736 | **48,821** | 116,658 | 46,412 | **70,246** |
| Lifetime QALYs | 7.03 | 6.70 | **0.33** | 4.43 | 2.95 | **1.48** |
| ICER | | | **£149,748** | | | **£47,503** |

M0 = non-metastatic at baseline; M1 = metastatic at baseline; AAP = abiraterone acetate plus prednisone/prednisolone; SOC = standard of care; QALYs = quality-adjusted life years; ICER = incremental cost-effectiveness ratio.

cohort is reported elsewhere [12]. Participants were followed up for a median of 3.08 years in the M0 subgroup and 3.00 years in the M1 subgroup (S4 File).

**Survival modelling.** Table 2 shows the numbers of times that each transition was used during the observed trial period. There were 9 models required in total: 6 for individual transitions and 3 for groups of jointly modelled transitions. The shaded groups indicate the three jointly modelled groups: time to first event (i.e. transitions from HS1/2/3 to HS4/5/7/8), time to other-cause death (from HS1-7 to HS9), and transitions to the worst CRPC state (HS7) from the other three CRPC states (HS4/5/6), which was performed jointly due to small event numbers. Numbers with borders indicate the six singly modelled transitions (HS4 to HS5, HS5 to HS6, HS4 to HS8, HS5 to HS8, HS6 to HS8, and HS7 to HS8).

All model parameters are given in S4 File. The results of the joint model describing the time to first event, or treatment failure, show that those in the AAP+SOC arm had a significantly longer time to first event than the SOC-only arm. The joint model considering the time to other-cause death depended on age, with the oldest patients having longer time to other-cause death compared to the youngest age group (an apparently contradictory but long-established finding [48]). Other transitions besides first event and other-cause death were also modelled controlling for randomised group, and relevant baseline covariates were assessed for inclusion in each model. Further details on these models and their validation is given in S4 File.

**Table 2. Transition matrix showing how many times each transition was used during the trial period.**

| from \ to | HS1 | HS2 | HS3 | HS4 | HS5 | HS6 | HS7 | HS8 | HS9 |
|---|---|---|---|---|---|---|---|---|---|
| HS1 | np | np | np | 194 | 14 | np | 7 | 4 | 29 |
| HS2 | np | np | np | np | 513 | np | 15 | 0 | 23 |
| HS3 | np | np | np | np | np | np | 34 | 2 | 1 |
| HS4 | np | np | np | np | **41** | np | 16 | **16** | 7 |
| HS5 | np | np | np | np | np | 124 | 40 | **150** | 16 |
| HS6 | np | np | np | np | np | np | 1 | **86** | 1 |
| HS7 | np | np | np | np | np | np | np | 63 | 6 |
| HS8 | np | np | np | np | np | np | np | np | np |
| HS9 | np | np | np | np | np | np | np | np | np |

HS1 (M0 subgroup), Naïve M0/M1a: M0, or M1 lymph nodes outside pelvis; HS2 (M1 subgroup), Naïve M1b: M1 bone mets; HS3 (M1 subgroup), Naïve M1c: M1 visceral mets; HS4, CRPC M0/M1a: M0, or M1 lymph nodes outside pelvis; HS5, CRPC M1b: M1 bone mets; HS6, CRPC SRE: M1 bone mets with SRE; HS7, CRPC M1c: M1 visceral mets; HS8, Prostate cancer death; HS9, Other cause death. SRE = skeletal-related events; mets = metastases; CRPC = castrate-resistant prostate cancer; M0 = non-metastatic at baseline; M1 = metastatic at baseline; np = not possible i.e. this transition is not possible according to Fig 1.

**Table 3. Mean daily undiscounted per-patient costs of AAP: Raw reported doses and imputed doses.**

| | Number of patients | Daily cost of reported dose | | | Daily cost of imputed dose | | | Treatment duration (days) | |
|---|---|---|---|---|---|---|---|---|---|
| | | n events | mean (£) | SD (£) | n events | mean (£) | SD (£) | mean | SD |
| **Abiraterone** | | | | | | | | | |
| AAP+SOC arm | 951 | 1154 | 94.69 | 10.37 | 1184 | 94.77 | 10.24 | 590.2 | 438.7 |
| SOC-only arm | 121 | 0 | - | - | 127 | 97.68 | 0.00 | 256.9 | 235.0 |
| **Prednisolone** | | | | | | | | | |
| AAP+SOC arm | 951 | 3 | 0.07 | 0.04 | 1186 | 0.02 | 0.00 | 589.0 | 439.2 |
| SOC-only arm | 122 | 1 | 0.09 | - | 128 | 0.02 | 0.01 | 255.2 | 234.8 |

Treatment duration was counted to censor date or earlier. There are more 'events' than patients because some patients stopped then re-started. SD = standard deviation; AAP = abiraterone acetate plus prednisone/prednisolone; SOC = standard of care.

**Utilities.** Of the 1,917 participants, 1,794 (898 AAP+SOC and 896 SOC-only) agreed to complete the EQ-5D-3L, providing 15,941 completed questionnaires. Despite the protocol not requiring EQ-5D-3L to be collected after progression, some patients did report this information, allowing utility scores to be modelled using trial data. The mean, standard deviation, range and graphical presentations of the raw, unimputed utility scores along with levels and patterns of missingness at each timepoint are provided in S5 File.

Five imputations were performed and analysed separately by arm. The imputation with the best fit for the two-part regression analysis described in the *Methods*: *Trial-based analysis*: *Utility scores* section was used to estimate utilities in the lifetime simulation model (see *Methods*: *Lifetime simulation model*: *Utility scores and quality-adjusted life-years* section for further details). The regression parameters and one-way deterministic sensitivity analysis varying these parameters are given in S5 File, followed by the imputed utility scores from the best-fit imputation used in the regression, and the predicted utility scores from the regression. These ranged from predicted means of 0.754 (SD 0.059) to 0.802 (SD 0.044) in the naïve health states (HS1-3) and 0.632 (SD 0.060) to 0.744 (SD 0.036) in the CRPC health states (HS4-7). Further details are given in S5 File.

**Cost information from trial data.** Mean daily costs per patient for investigational drugs (AAP) are given in Table 3. The two-part regression for general disease management was split into costs above and below £1,500 per 42-day cycle, including a generalised gamma model for the lower costs, and a Gaussian model for the higher costs. Further detailed information including the regression model outputs is given in S6 File.

## Lifetime simulation model

**Survival and quality-adjusted survival.** The model predicted overall survival increased from 4.97 (SOC-only) to 7.65 years (AAP+SOC), and discounted quality-adjusted survival increased from 2.95 (SOC-only) to 4.43 QALYs (AAP+SOC), for M1 patients. The corresponding increases for M0 patients were from 12.46 (SOC-only) to 12.75 years (AAP+SOC), and 6.70 (SOC-only) to 7.03 QALYs (AAP+SOC). Regarding patients' first events, the model predicted failure-free survival increased from 2.64 (SOC-only) to 6.43 years (AAP+SOC) and discounted quality-adjusted failure-free survival increased from 1.65 (SOC-only) to 3.79 QALYs (AAP+SOC), for M1 patients. The corresponding increases for M0 patients were from 8.20 (SOC-only) to 12.14 years (AAP+SOC) and from 4.72 (SOC-only) to 6.72 QALYs (AAP +SOC). The effect of adding AAP to SOC was therefore to extend overall and failure-free survival and quality-adjusted survival and seemed to delay progression to worse disease states for

M1 patients. These results are shown for the M0 (HS1) and M1 (split according to the type of metastasis according to HS2 (bone) and HS3 (visceral)) patient subgroups in Fig 2, and for the overall patient population in S7 File.

**Probabilistic sensitivity analysis.**   The cost-effectiveness plane (CEP) (top of Fig 3) generated using probabilistic methods shows that AAP+SOC treatment pathway is both more expensive and more effective than SOC-only pathway in the M1 subgroup. Cost-effectiveness acceptability curves (CEAC) were plotted for each subgroup and suggest a 12.0% chance of AAP+SOC being cost-effective compared to SOC-only for a threshold of £30,000/QALY in the M1 subgroup (bottom of Fig 3), and a 2.4% chance in the M0 subgroup (S9 File).

**Deterministic abiraterone cost threshold analysis.**   The BNF cost of abiraterone acetate was likely to be an over-estimate of the price paid by the NHS. For the calculated ICER to fall below the £30,000/QALY threshold, the cost of abiraterone acetate would need to be below 63% (£62/day) of the current BNF price for M1 patients, and below 29% (£28/day) of the BNF price for M0 patients. If the cost of abiraterone acetate were below 11.7% (£11/day) of the current BNF price, these results suggested that use of AAP+SOC would dominate the SOC-only pathway for M0 patients, i.e. cost less and provide more QALYs. A similar cost-saving scenario is not available for M1 patients in our model because the break-even point for zero incremental cost in this group is at a negative price for AAP (S10 File).

## Discussion

The survival results from this analysis corroborated those of the clinical effectiveness paper [22] in suggesting there was both a survival benefit from addition of AAP to SOC, and a delay in progression to worse disease states. However, using BNF pricing for abiraterone acetate, there was a 12.0% probability for M1 patients, and 2.4% probability for M0 patients, that AAP was cost-effective compared to SOC-only, at a threshold of £30,000 per QALY gained for hormone-naïve men with prostate cancer, suggesting that addition of abiraterone acetate to ADT was not cost-effective in this context, despite being effective in improving survival and discounted quality-adjusted survival for these patients.

Our results, which used patient-level data directly from the STAMPEDE study in a lifetime decision model to avoid bias due to short trial follow-up, are similar to those reported by other groups who performed secondary cost-utility analyses using published data from STAMPEDE and other trials in similar populations. Sathianathen et al. reported a three-arm cost-effectiveness analysis comparing ADT, ADT+docetaxel and ADT+AAP, based on aggregated data from the STAMPEDE trial, and concluded that addition of AAP was likely to be the most effective option in terms of increasing QALYs, but would not become the most cost-effective unless the monthly price of abiraterone was reduced below $3114 (2017 US $, using cost-effectiveness threshold of $100,000/QALY gained) [17]. A network meta-analysis comparing a number of strategies in this population concluded that ADT+AAP was likely to be the most effective [18], and a further model-based analysis by Sung et al. considered a series of comparisons between ADT, ADT+docetaxel, ADT+AAP, ADT+enzalutamide and ADT+apalutamide, and illustrated a five-way cost-effectiveness frontier [42] that suggested that ADT+AAP would be the preferred option at a cost-effectiveness threshold of $100,000/QALY gained, although ADT alone or ADT+docetaxel would be preferred at lower thresholds [49].

This lifetime model used data with a median trial follow-up of around 3 years to extend the model to a 45-year horizon. The parametric survival models predicted the short-term trial survival curves well, replicating the results of the trial, and longer-term predictions made by the model were validated by comparison to other published work (see S4 File), although only limited information was available for this validation due to standard short follow-up in clinical

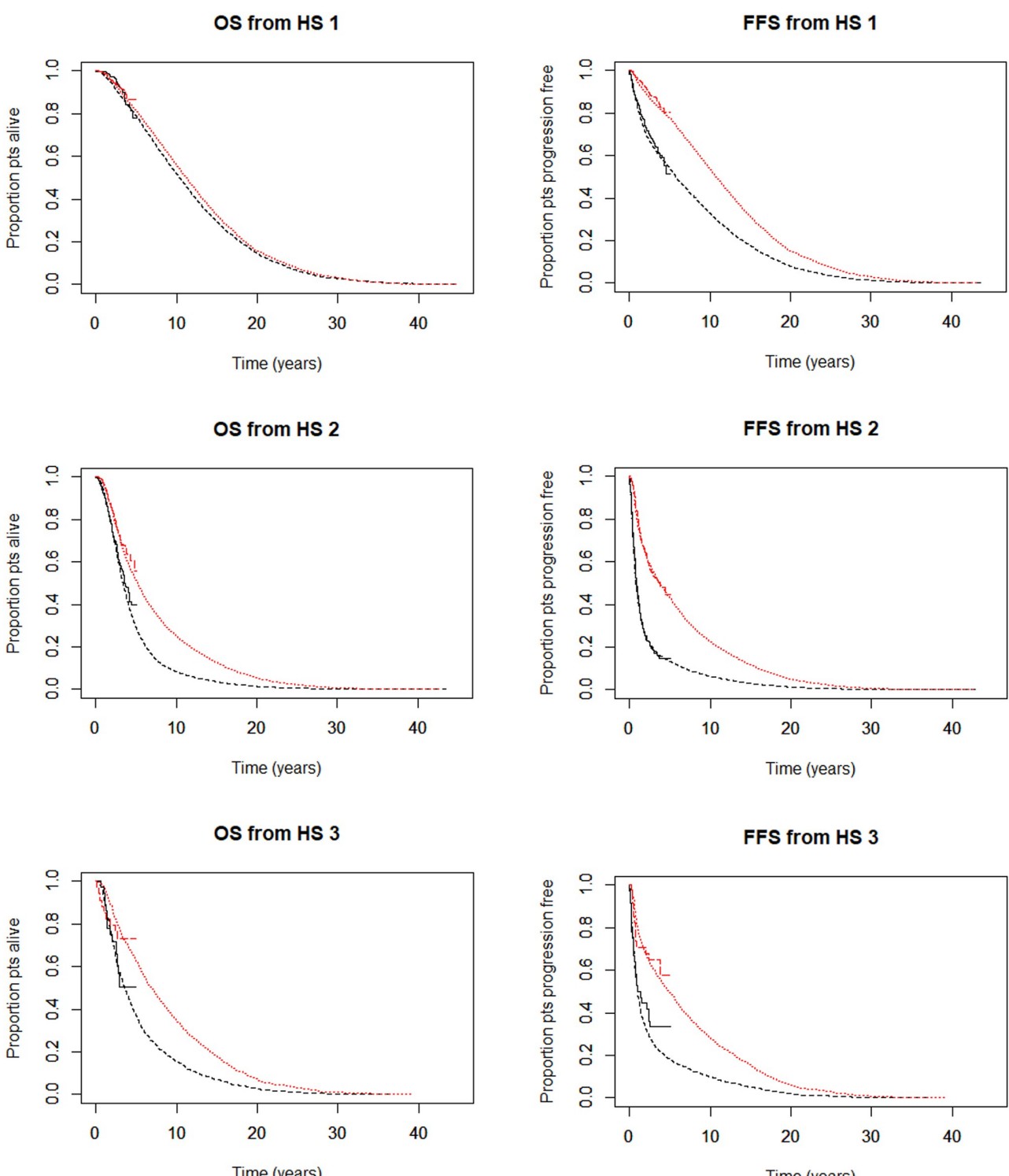

**Fig 2. Overall and failure-free survival from each of the three starting health states.** Top row: Overall (OS) and failure-free survival (FFS) from HS1, i.e. the M0 subgroup; middle row: OS and FFS from HS2, i.e. M1 patients whose initial metastasis was in bone; bottom row: OS and FFS from HS3, i.e. M1 patients whose initial metastasis was in visceral tissue (not bone). Stepped lines to around 5 years are observed trial data and smooth lines to 45 years are predicted data from the lifetime simulation model. In the lifetime lines, short dashes (red) indicate the AAP+SOC arm, and longer dashes (black) indicate the SOC-only arm.

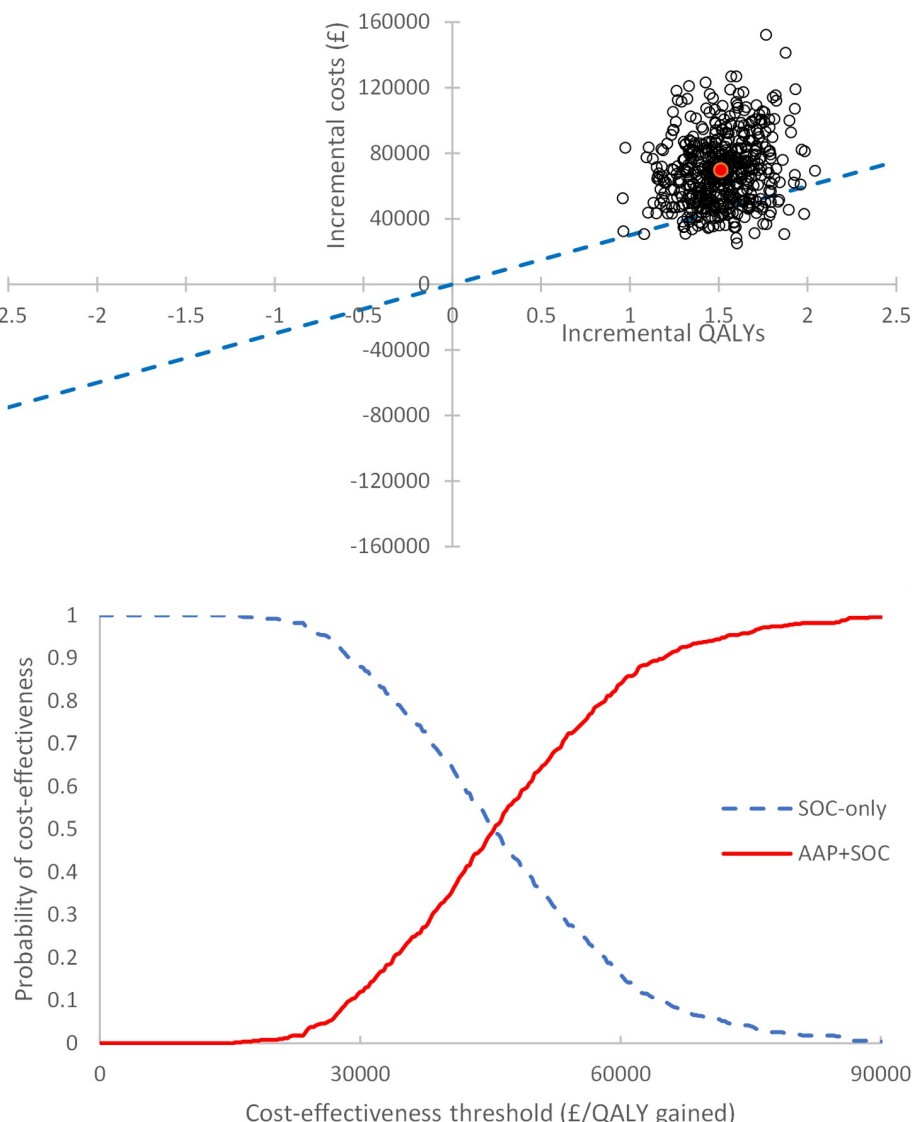

**Fig 3. Cost-effectiveness plan and cost-effectiveness acceptability curve for the M1 subgroup.** Top: Cost-effectiveness plane (CEP) for M1 subgroup, using base case BNF price for abiraterone. The red point indicates the mean incremental cost plotted against the mean incremental QALYs in this set of probabilistic results and the blue dotted line indicates the £30,000/QALY gained cost-effectiveness threshold. Bottom: Cost-effectiveness acceptability curve (CEAC) for M1 subgroup, using base-case BNF price for abiraterone.

trials. A further limitation of the analysis was that STAMPEDE trial data were not complete regarding medications or disease progression events, as complete follow-up post-progression was not mandatory. These gaps were filled for the M0 subgroup by assuming that outcomes after metastases mimicked those for M1 patients, and clinical experts felt that the corresponding gaps for M1 patients were not likely to be substantial. There was less quality-of-life information reported in progressed disease, as quality-of-life data (EQ-5D-3L) were not routinely collected after disease progression, but that which was obtained was considered to more adequately describe the experience of this population than using literature values from other studies.

Regarding contextual limitations, changes were made to SOC during the study, which led to 121 of the 957 patients in the SOC-only arm also receiving AAP at some stage during the study, mostly during later disease stages. This number could have grown in the time since the dataset was frozen. Many STAMPEDE patients were yet to report further treatment for prostate cancer, and patterns of secondary treatments may change over time. Further changes to SOC occurred after this section of the STAMPEDE study was completed, so neither arm in this cost-effectiveness analysis exactly replicates current UK practice; in particular, docetaxel in addition to ADT is increasingly offered earlier in the pathway, to patients newly presenting with metastatic disease, with around 27% of this group taking this up in 2019 [50]. Further pathway changes took place in 2020 as a result of the pandemic, which are not reflected in this analysis. Subgroup analysis to examine more specific treatment pathways, besides the M0/M1 split, could not be undertaken due to small patient numbers. Nevertheless, sensitivity analyses conducted to assess the impact of uncertainties on the lifetime model results suggested the overall cost-effectiveness conclusions were robust to these uncertainties, and the key cost-effectiveness driver remained the price of abiraterone acetate.

The likelihood of the addition of AAP being cost-effective depends heavily on its price and also on the comparator. This work focuses on ADT monotherapy, however, around 20% of men receive upfront docetaxel in addition but no men within part of STAMPEDE received this therapy so we are unable to provide data for this comparator. The abiraterone acetate patent is expected to expire in England in September 2022, so generic manufacturers will then be allowed to enter the market, which is likely to lead to lower prices. Healthcare pathway costs in health economic evaluation are a proxy for opportunity cost, the economic concept at the heart of economic evaluation. Use of the commonly quoted NICE cost-effectiveness thresholds of £20,000 to £30,000 per QALY gained may lead to a net loss in health across the system because research suggests that the average cost of producing a QALY is £13,000 in the NHS in England. As a result, adoption of a new therapy with an ICER greater than £13,000/QALY gained would produce net fewer QALYs than current standard of care [15], and the higher the ICER, the greater this net loss of health. Although AAP+SOC is clearly effective compared to SOC, at current BNF prices of more than £35,000 per patient per year for AAP, it would not meet the conditions required to represent value for money to the English NHS without a discount on this price. The NHS purchases abiraterone acetate at an undisclosed discount so we do not know whether the current price meets cost-effectiveness thresholds. These results apply to the English context, as other countries' decision-making bodies use cost-per-QALY analysis to differing degrees and with different thresholds when considering approvals of new therapies, and the cost of generating a QALY would also be different in different countries.

As well as considering price discounts, there has also been work published in recent years on the impact of using a lower dose of abiraterone acetate (250mg/day) with a low-fat meal, as AAP was delivered under fasting conditions in its pivotal trials, which would not have taken advantage of the large food effect [51, 52]. A reduction in dose by 75% would have a similar impact on cost as a reduction in price by 75%, so if taking AAP with food does indeed not reduce its effectiveness, then the lower dose would be cost-effective in both subgroups according to the threshold analysis described above (see also S10 File).

## Conclusions

Our results find a low probability for AAP being cost-effective in the English NHS at the BNF price as a first-line treatment alongside hormone therapy for patients with non-metastatic and metastatic disease, although it has the potential to be cost-effective, or even cost-saving in M0 patients, with lower pricing according to the thresholds described above. As the NHS

purchases abiraterone acetate at an undisclosed discount, it is impossible to assess whether in fact it is already being bought at a price that meets standard NICE cost-effectiveness metrics. Regarding AAP's current indication in the UK, we note that the marketing authorisation covers only a proportion of those in STAMPEDE's M1 subgroup [13], based on the LATITUDE study [5], so an expansion of this authorisation would also be required. A NICE appraisal relating to the licensed indication for M1 patients recently concluded that in the licensed population, the treatment was not cost-effective in comparison to ADT plus docetaxel, despite the fact that only a minority of men received this combination, while the majority (pre-pandemic) received ADT alone [53].

## Supporting information

**S1 Checklist.**
(PDF)

**S1 File.**
(PDF)

**S2 File.**
(PDF)

**S3 File.**
(PDF)

**S4 File.**
(PDF)

**S5 File.**
(PDF)

**S6 File.**
(PDF)

**S7 File.**
(PDF)

**S8 File.**
(PDF)

**S9 File.**
(PDF)

**S10 File.**
(PDF)

## Acknowledgments

The authors are very grateful to all patients participating in the STAMPEDE study and their families, and to all the staff at sites and otherwise involved in the research, including Claire Amos who manages the study. Lists of trial oversight committees and investigators are given in James et al. [12]. The authors are also very grateful to Beth Woods for her invaluable input in terms of providing the initial code which was adapted for this analysis, and her insights into conducting the analysis. The authors also wish to thank the thoughtful and constructive journal peer reviewers who have provided their valuable insights to improve this manuscript.

## Author Contributions

**Conceptualization:** Caroline S. Clarke, Rachael M. Hunter, David P. Dearnaley, David Matheson, Gerhardt Attard, Hannah L. Rush, Rob J. Jones, William Cross, Chris Parker, J. Martin Russell, Robin Millman, Silke Gillessen, Zafar Malik, Jason F. Lester, James Wylie, Noel W. Clarke, Mahesh K. B. Parmar, Matthew R. Sydes, Nicholas D. James.

**Data curation:** Christopher D. Brawley, Fiona C. Ingleby.

**Formal analysis:** Caroline S. Clarke, Rachael M. Hunter, Andrea Gabrio.

**Funding acquisition:** Caroline S. Clarke, Rachael M. Hunter, Nicholas D. James.

**Investigation:** David P. Dearnaley, David Matheson, Gerhardt Attard, Hannah L. Rush, Rob J. Jones, William Cross, Chris Parker, J. Martin Russell, Robin Millman, Silke Gillessen, Zafar Malik, Jason F. Lester, James Wylie, Noel W. Clarke, Mahesh K. B. Parmar, Matthew R. Sydes, Nicholas D. James.

**Methodology:** Matthew R. Sydes, Nicholas D. James.

**Project administration:** Matthew R. Sydes, Nicholas D. James.

**Supervision:** Rachael M. Hunter.

**Validation:** Caroline S. Clarke, Rachael M. Hunter, Andrea Gabrio.

**Writing – original draft:** Caroline S. Clarke.

**Writing – review & editing:** Caroline S. Clarke, Rachael M. Hunter, Andrea Gabrio, Christopher D. Brawley, Fiona C. Ingleby, David P. Dearnaley, David Matheson, Gerhardt Attard, Hannah L. Rush, Rob J. Jones, William Cross, Chris Parker, J. Martin Russell, Robin Millman, Silke Gillessen, Zafar Malik, Jason F. Lester, James Wylie, Noel W. Clarke, Mahesh K. B. Parmar, Matthew R. Sydes, Nicholas D. James.

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
