## [Decision Letter · Decision Letter 0]

21 Feb 2022

PONE-D-22-01674Cost-utility analysis of adding abiraterone acetate plus prednisone/prednisolone to long-term hormone therapy in newly diagnosed advanced prostate cancer in England: lifetime decision model based on STAMPEDE trial dataPLOS ONE

Dear Dr. Clarke,

Thank you for submitting your manuscript to PLOS ONE. After careful consideration, we feel that it has merit but does not fully meet PLOS ONE’s publication criteria as it currently stands. Therefore, we invite you to submit a revised version of the manuscript that addresses the points raised during the review process.

We look forward to receiving your revised manuscript.

Kind regards,

Giandomenico Roviello

Academic Editor

PLOS ONE

Journal Requirements:

1. Please ensure that your manuscript meets PLOS ONE's style requirements, including those for file naming. The PLOS ONE style templates can be found at https://journals.plos.org/plosone/s/file?id=wjVg/PLOSOne_formatting_sample_main_body.pdf and https://journals.plos.org/plosone/s/file?id=ba62/PLOSOne_formatting_sample_title_authors_affiliations.pdf. 2. Our staff editors have determined that your manuscript may be within the scope of our Cancer and Social Inequity Call for Papers. This editorial initiative is headed by a team of Guest Editors for PLOS ONE: Vesna Zadnik (Institute of Oncology, Ljubljana), Nixon Niyonzima (Uganda Cancer Institute), Claudia Allemani (London School of Hygiene and Tropical Medicine). This call for papers aims to highlight the negative impacts of social inequities on health, identify the effects of social and corporate policies on access to healthcare services, and propose solutions to promote more equitable cancer outcomes and ultimately, social justice.  Additional information can be found on our announcement page: https://collections.plos.org/call-for-papers/cancer-and-social-inequality/If you would like your manuscript to be considered for this collection, please let us know in your cover letter and we will ensure that your paper is treated as if you were responding to this call.  Please note that being considered for the Collection does not require additional peer review beyond the journal’s standard process and will not delay the publication of your manuscript if it is accepted by PLOS ONE. If you would prefer to remove your manuscript from collection consideration, please specify this in the cover letter. 3. Thank you for stating the following in the Competing Interests section: [Besides the funding declared above,•
CDB reports grants from Novartis, Sanofi-Aventis, Pfizer, Janssen Pharma, Cancer Research UK, and Medical Research Council.•
DD reports personal fees from The Institute of Cancer Research, grants from Cancer Research UK Program Grant and personal fees from Janssen. In addition, DD has a patent EP1933709B1 issued.•
GA reports receiving commercial research grants from Janssen and AstraZeneca; has received honoraria and/or travel support from the speakers’ bureaus of Janssen, Astellas, Pfizer, Ferring, Sanofi-Aventis and Roche/Ventana; and has served as a consultant for/advisory board member of Janssen, Bayer, Astellas, Pfizer, Novartis, AstraZeneca, Orion, and Essa. GA has an ownership interest (including patents) in The Institute of Cancer Research Rewards to Discoverers for abiraterone acetate. •
RJJ reports grants and personal fees from Astellas, AstraZeneca, Exelixis, and Roche; grants, personal fees and non-financial support from Bayer; personal fees and non-financial support from Bristol Myers Squibb, Janssen, Ipsen, and MSD; and personal fees from Merck Serono, Novartis, Pfizer, Sanofi Genzyme, and EUSA.•
WC reports grants from Myriad Genetics.•
CP reports personal fees from Bayer, Clarity, ITM (Isotopen Technologien Muenchen AG), Janssen, and Myovant.•
SG reports personal fees from Sanofi, Orion, Roche, Janssen Cilag, and Amgen; other benefits from Menarini Silicon Biosystems, Bayer, AAA International, ProteoMediX, Toledo, and MSD; personal fees and other benefits from Astellas Pharma; and grants from Astellas Pharma.•
ZM reports involvement in consultancy and advisory boards at Janssen and Sanofi, in advisory boards at Astellas, and sponsorship to attend medical conferences from Astellas, Bayer and Janssen.•
NWC reports receiving research grants AstraZeneca and Janssen; honoraria and/or travel support from the speakers’ bureaus of Janssen, Astellas, Ferring, Sanofi-Aventis and has served as a consultant for/advisory board member of Janssen, Bayer, Astellas, Ferring, and AstraZeneca. •
MKBP reports unrestricted grant funding to contribute to STAMPEDE overall from Astellas, Clovis Oncology, Novartis, Pfizer and Sanofi.•
MRS reports grants from Clovis, grants and non-financial support from Astellas, Janssen, Novartis, Pfizer, and Sanofi to support the running of STAMPEDE; and personal fees from Lilly Oncology and Janssen for educational events unconnected to the submitted work or the underpinning trial.•
NDJ reports grants and personal fees from Janssen, Astellas, and Sanofi, and personal fees from AstraZeneca, during the conduct of the study.•
CSC, RMH, AG, FCI, DM, HLR, JMR, RM, JFL and JW have nothing to disclose.]  Please confirm that this does not alter your adherence to all PLOS ONE policies on sharing data and materials, by including the following statement: "This does not alter our adherence to  PLOS ONE policies on sharing data and materials.” (as detailed online in our guide for authors http://journals.plos.org/plosone/s/competing-interests).  If there are restrictions on sharing of data and/or materials, please state these. Please note that we cannot proceed with consideration of your article until this information has been declared.  Please include your updated Competing Interests statement in your cover letter; we will change the online submission form on your behalf. 4. In your Data Availability statement, you have not specified where the minimal data set underlying the results described in your manuscript can be found. PLOS defines a study's minimal data set as the underlying data used to reach the conclusions drawn in the manuscript and any additional data required to replicate the reported study findings in their entirety. All PLOS journals require that the minimal data set be made fully available. For more information about our data policy, please see http://journals.plos.org/plosone/s/data-availability. Upon re-submitting your revised manuscript, please upload your study’s minimal underlying data set as either Supporting Information files or to a stable, public repository and include the relevant URLs, DOIs, or accession numbers within your revised cover letter. For a list of acceptable repositories, please see http://journals.plos.org/plosone/s/data-availability#loc-recommended-repositories. Any potentially identifying patient information must be fully anonymized. Important: If there are ethical or legal restrictions to sharing your data publicly, please explain these restrictions in detail. Please see our guidelines for more information on what we consider unacceptable restrictions to publicly sharing data: http://journals.plos.org/plosone/s/data-availability#loc-unacceptable-data-access-restrictions. Note that it is not acceptable for the authors to be the sole named individuals responsible for ensuring data access. We will update your Data Availability statement to reflect the information you provide in your cover letter. 5. One of the noted authors is a group or consortium [the STAMPEDE investigators]. In addition to naming the author group, please list the individual authors and affiliations within this group in the acknowledgments section of your manuscript. Please also indicate clearly a lead author for this group along with a contact email address. 6. Your ethics statement should only appear in the Methods section of your manuscript. If your ethics statement is written in any section besides the Methods, please delete it from any other section. 

Reviewers' comments:

Reviewer's Responses to Questions

**Comments to the Author**

1. Is the manuscript technically sound, and do the data support the conclusions?

Reviewer #1: Yes

Reviewer #2: Yes

2. Has the statistical analysis been performed appropriately and rigorously? 

Reviewer #1: Yes

Reviewer #2: I Don't Know

3. Have the authors made all data underlying the findings in their manuscript fully available?

Reviewer #1: Yes

Reviewer #2: Yes

4. Is the manuscript presented in an intelligible fashion and written in standard English?

Reviewer #1: Yes

Reviewer #2: Yes

5. Review Comments to the Author

Reviewer #1: The authors present results of a cost-utility analysis of adding abiraterone acetate and prednisone or prednisolone (AAP) to long-term hormone therapy in men newly diagnosed with advanced prostate cancer. They created a simulation model to simulate time in various states based on STAMPEDE trial data. This information was then used in combination with lifetime costs and quality adjusted life years also estimated using trial data, along with data from the literature when needed to assess the cost-effectiveness of the treatment. Authors concluded that AAP was not cost-effective at the current estimated cost, but also present scenarios where it could be cost-effective. Authors provide detailed explanations of the methods used (and results to support decisions in supplemental material). There are a couple of minor comments that authors should address.

1. on page 8, authors state that the dose information for abiraterone in the SOC-only arm was missing and imputed as the indicated and modal observed amount. Was this for those individuals that later took AAP after it was approved for certain individuals later during the trial? I was confused the first time reading this, since it was not clear why those in the SOC-only arm would be getting abiraterone. Authors should consider adding a clarifying statement or phrase at this part of the paper to remind readers that the SOC-only arm participants could have had AAP later during the course of their treatment (or if that is not the explanation, clarification on why they had AAP).

2. on page 11, in the paragraph just before Section 3.2, "and the goes on to describe" should be "and then goes on to describe"

Reviewer #2: The present article excels in approaching a drug whose clinical benefit has already been demonstrated in the literature (especially STAMPEDE, but also in other studies) but which has a high cost. The adoption of new health policies and the introduction of new medications must, in fact, take into account the economic component: public health is chronically lacking financial resources globally, and this pressure has been heightened in the context of the COVID-19 pandemic that we still meet.

Some doubts we observed during the study:

-It is reiterated several times throughout the article that the exact amount of abiraterone acetate paid by the UK healthcare system is not known.

-With the use of the algorithm developed, it was possible to reach a value, at least for patients in stage M0, in which its use would be economically viable, but as the real value paid is unknown, it is not possible to know if values similar to these are practiced.

-The study is largely based on the STAMPEDE study, taking into account projections stipulated by it, so it is at the mercy of its limitations and its own biases.

-Does not explain why the ICER (incremental cost-effectiveness ratio) was higher for subgroup M1 than M0.

The breach of the medication patent and studies that make the applied dose more flexible are among the changes in the scenario that can make the economic aspect more favorable to the wide adoption of the use of abiraterone acetate in the public health system.

6. PLOS authors have the option to publish the peer review history of their article (what does this mean?). If published, this will include your full peer review and any attached files.

Reviewer #1: No

Reviewer #2: No

---

## [Author Response · Author response to Decision Letter 0]

26 Apr 2022

From: Editor

• Thanks, we have fixed this.

2. Our staff editors have determined that your manuscript may be within the scope of our Cancer and Social Inequity Call for Papers. This editorial initiative is headed by a team of Guest Editors for PLOS ONE: Vesna Zadnik (Institute of Oncology, Ljubljana), Nixon Niyonzima (Uganda Cancer Institute), Claudia Allemani (London School of Hygiene and Tropical Medicine). This call for papers aims to highlight the negative impacts of social inequities on health, identify the effects of social and corporate policies on access to healthcare services, and propose solutions to promote more equitable cancer outcomes and ultimately, social justice. Additional information can be found on our announcement page: https://collections.plos.org/call-for-papers/cancer-and-social-inequality/

If you would like your manuscript to be considered for this collection, please let us know in your cover letter and we will ensure that your paper is treated as if you were responding to this call. Please note that being considered for the Collection does not require additional peer review beyond the journal’s standard process and will not delay the publication of your manuscript if it is accepted by PLOS ONE. If you would prefer to remove your manuscript from collection consideration, please specify this in the cover letter.

• Thanks, we would be delighted to be included in this collection and have noted this in the cover letter.

[...]

• Thanks, we have added this sentence to the COI section and included it in the cover letter, and clarified the data sharing section as described below.

• Thanks, we have clarified the data sharing statement in the cover letter. The STAMPEDE study involves human research participants and contains sensitive information. We therefore will make the dataset available on request via our established MRC CTU at UCL processes, described here: (https://www.mrcctu.ucl.ac.uk/our-research/other-research-policy/data-sharing/). 

5. One of the noted authors is a group or consortium [the STAMPEDE investigators]. In addition to naming the author group, please list the individual authors and affiliations within this group in the acknowledgments section of your manuscript. Please also indicate clearly a lead author for this group along with a contact email address.

• Thanks, we have amended this. The STAMPEDE investigators is a very large group, so instead of adding directly to the paper we have referred to an online appendix from another paper for the list of names and affiliations, and have added a contact email address for this group mrcctu.stampede@ucl.ac.uk. We have also mentioned this in the cover letter.

• Thanks, we have fixed this.

• Thanks, we have checked this and tidied up some duplicates – apologies for that. We did not find any that had been retracted (using EndNote 20’s automatic function) – please do let me know of any specific papers that you think have been retracted and I will check again.

• I have also added in missing references in supplementary files: Ramsay et al., and Stangelberger et al.

• And added in a new reference to a recently published NICE technology appraisal (TA721) – this is also mentioned in the cover letter.

From: Reviewer #1

Reviewer #1: The authors present results of a cost-utility analysis of adding abiraterone acetate and prednisone or prednisolone (AAP) to long-term hormone therapy in men newly diagnosed with advanced prostate cancer. They created a simulation model to simulate time in various states based on STAMPEDE trial data. This information was then used in combination with lifetime costs and quality adjusted life years also estimated using trial data, along with data from the literature when needed to assess the cost-effectiveness of the treatment. Authors concluded that AAP was not cost-effective at the current estimated cost, but also present scenarios where it could be cost-effective. Authors provide detailed explanations of the methods used (and results to support decisions in supplemental material). There are a couple of minor comments that authors should address.

1. on page 8, authors state that the dose information for abiraterone in the SOC-only arm was missing and imputed as the indicated and modal observed amount. Was this for those individuals that later took AAP after it was approved for certain individuals later during the trial? I was confused the first time reading this, since it was not clear why those in the SOC-only arm would be getting abiraterone. Authors should consider adding a clarifying statement or phrase at this part of the paper to remind readers that the SOC-only arm participants could have had AAP later during the course of their treatment (or if that is not the explanation, clarification on why they had AAP).

• Thanks for this very useful comment. We have clarified this in the revised paper (yes, the reason was what the reviewer suggested).

2. on page 11, in the paragraph just before Section 3.2, "and the goes on to describe" should be "and then goes on to describe"

• Thanks, we have fixed this.

From: Reviewer #2

Reviewer #2: The present article excels in approaching a drug whose clinical benefit has already been demonstrated in the literature (especially STAMPEDE, but also in other studies) but which has a high cost. The adoption of new health policies and the introduction of new medications must, in fact, take into account the economic component: public health is chronically lacking financial resources globally, and this pressure has been heightened in the context of the COVID-19 pandemic that we still meet. 

Some doubts we observed during the study:

-It is reiterated several times throughout the article that the exact amount of abiraterone acetate paid by the UK healthcare system is not known.

• Yes, this is correct. This information is commercially sensitive and the manufacturers do not disclose this information, so we do not know it.

-With the use of the algorithm developed, it was possible to reach a value, at least for patients in stage M0, in which its use would be economically viable, but as the real value paid is unknown, it is not possible to know if values similar to these are practiced.

• Yes, this is correct. We do not know the current price as the manufacturers have not disclosed it to us.

-The study is largely based on the STAMPEDE study, taking into account projections stipulated by it, so it is at the mercy of its limitations and its own biases.

• Yes, this is correct and is mentioned throughout in the paper at appropriate points. We would be happy to include further discussion on this if that would be useful.

-Does not explain why the ICER (incremental cost-effectiveness ratio) was higher for subgroup M1 than M0.

• These two patient subgroups had different lifetime costs and QALYs in each of the AAP and SOC arms, and the differences between the arms for the costs and QALYs were also different. Table 1 and section 3.3.1 address these differences.

The breach of the medication patent and studies that make the applied dose more flexible are among the changes in the scenario that can make the economic aspect more favorable to the wide adoption of the use of abiraterone acetate in the public health system.

• No response required.

---

## [Editor Report · Decision Letter 1]

17 May 2022

Cost-utility analysis of adding abiraterone acetate plus prednisone/prednisolone to long-term hormone therapy in newly diagnosed advanced prostate cancer in England: lifetime decision model based on STAMPEDE trial data

PONE-D-22-01674R1

Dear Dr. Clarke,

We’re pleased to inform you that your manuscript has been judged scientifically suitable for publication and will be formally accepted for publication once it meets all outstanding technical requirements.

Kind regards,

Giandomenico Roviello

Academic Editor

PLOS ONE